# A strong fracture-resistant high-entropy alloy with nano-bridged honeycomb microstructure intrinsically toughened by 3D-printing

Punit Kumar [1,2,7], Sheng Huang [3,7], David H. Cook [1,2,7], Kai Chen [4], Upadrasta Ramamurty [3,5], Xipeng Tan [6] ✉ & Robert O. Ritchie [1,2] ✉

Strengthening materials via conventional "top-down" processes generally involves restricting dislocation movement by precipitation or grain refinement, which invariably restricts the movement of dislocations away from, or towards, a crack tip, thereby severely compromising their fracture resistance. In the present study, a high-entropy alloy $Al_{0.5}CrCoFeNi$ is produced by the laser powder-bed fusion process, a "bottom-up" additive manufacturing process similar to how nature builds structures, with the microstructure resembling a nano-bridged honeycomb structure consisting of a face-centered cubic (*fcc*) matrix and an interwoven hexagonal net of an ordered body-centered cubic B2 phase. While the B2 phase, combined with high-dislocation density and solid-solution strengthening, provides strength to the material, the nano-bridges of dislocations connecting the *fcc* cells, *i.e.*, the channels between the B2 phase on the cell boundaries, provide highways for dislocation movement away from the crack tip. Consequently, the nature-inspired microstructure imparts the material with an excellent combination of strength and toughness.

Yield strength, $\sigma_y$, and fracture toughness, $K_{JIc}$, are mutually exclusive properties in many structural materials[1,2]. The trade-off between these properties is particularly strong in the toughest materials, i.e., metals and their alloys. For example, CrCoNi-based medium/high-entropy alloys and cryogenic steels with face-centered cubic (*fcc*) crystal structures invariably are extremely ductile, i.e., they have a low resistance to plastic deformation[3–7]. The fracture resistance of these alloys is driven by plasticity ahead of the crack tip, which requires a relatively low yield strength and extended strain hardening[3,4]. During plastic deformation, a crack acts as a source and sink of dislocations, which

effectively blunts and shields the crack, which allows it to resist propagation, i.e., the intrinsic toughness of the material results in an excellent crack-initiation toughness ($K_{JIc}$)[2]. Increasing the strength of these materials by conventional alloy design generally involves restricting the movement of dislocations by precipitation hardening or grain refinement, which also restricts dislocation movement away from, or towards, the crack tip, such that the blunting or shielding effects are significantly reduced, often resulting in lower fracture resistance[1,2]. In some cases, grain refinement can improve the intrinsic toughness of the material if grain boundaries act as both source and

[1]Department of Materials Science and Engineering, University of California, Berkeley, CA, USA. [2]Materials Sciences Division, Lawrence Berkeley National Laboratory, Berkeley, CA, USA. [3]School of Mechanical and Aerospace Engineering, Nanyang Technological University, Singapore, Singapore. [4]Center for Advancing Materials Performance from the Nanoscale (CAMP-Nano), State Key Laboratory for Mechanical Behavior of Materials, Xi'an Jiaotong University, Xi'an, China. [5]Institute of Materials Research and Engineering, Agency for Science, Technology and Research (A*STAR), Singapore, Singapore. [6]Department of Mechanical Engineering, National University of Singapore, Singapore, Singapore. [7]These authors contributed equally: Punit Kumar, Sheng Huang, David H. Cook. ✉e-mail: xptan@nus.edu.sg; roritchie@lbl.gov

sink of dislocations. While improving fracture toughness by grain refinement is not universal, controlling the grain boundary's characteristics to improve an alloy's fracture toughness is also challenging[8–10]. In relatively high strength and low fracture toughness materials, the fracture resistance can be improved by extrinsic toughening mechanisms such as crack bridging, wedging, closure, and crack-path tortuosity[11–13]. But these extrinsic toughening mechanisms primarily act in the wake of a crack, where the resistance to crack growth only arises during the crack propagation[1]. Therefore, for many safety-critical applications, materials with high intrinsic crack-initiation toughness, $K_{JIc}$, are preferred as opposed to materials that derive their resistance to fracture during the crack growth, even when their lower strength is challenging to design for lightweight applications[1,11]. This highlights the lack of capability for microstructural design in conventional "top-to-down" processing to strengthen materials without affecting their intrinsic toughness. However, in the present study using the "bottom-up" additive manufacturing process of laser powder bed fusion (L-PBF), we develop a high-entropy $Al_{0.5}CrCoFeNi$ alloy with microstructure resembling a nano-bridged honeycomb of $fcc$ and body-centered cubic ($bcc$) phases, which generates an excellent combination of yield strength, $\sigma_y$, and crack-initiation fracture toughness, $K_{JIc}$, primarily driven by intrinsic toughening.

## Results
### Microstructure
Blocks of the $Al_{0.5}CrCoFeNi$ alloy (with a composition shown in the Methods section) were produced by the L-PBF process, which imparts a hierarchical microstructure consisting of meltpool boundaries, cellular structures, and nanoprecipitates[14,15]. The laser scan tracks or meltpool boundaries form due to the line-by-line and layer-by-layer printing processes (Supplementary Fig. 1a). The plane parallel to the build direction (BD) shows columnar grains of size (area weighted equivalent circle diameter) $\sim451 \pm 7\,\mu m$ (Supplementary Fig. 1), with a primarily <111> and <110> texture (Supplementary Fig. 1b). During additive manufacturing, the solidification microstructure is controlled by the ratio of thermal gradient, $G$, and cooling rate, $R$ (Supplementary Fig. 2a)[16]. The cooling rate, $R$, during the L-PBF process, is of order $\sim10^5$–$10^7$ K/s, and the thermal gradient, $G$, is $10^5$–$10^7$ K.m$^{-1}$;[17] therefore, the additively manufactured alloys generally have a microstructure consisting of a solidification cellular structure[15,18,19]. During cellular solidification, solutes with an equilibrium partition coefficient less than unity segregate onto the cell boundaries[16]. Such segregation is driven by the constitutional supercooling (ratio of $G$ and $R$) and the surface tension of the solute-rich liquid[18,20,21]. The solute-rich cell boundaries, after solidification, can also trap dislocations nucleated by thermal cycling during the L-PBF process and form a dislocation cellular structure[22–24]. In the case of the $Al_{0.5}CrCoFeNi$ alloy, the CrCoFe-rich $fcc$ matrix phase rejects Ni and Al onto the cell boundaries, since the equilibrium partition coefficients of Ni and Al are $\sim0.98$ and $\sim0.81$, respectively (Supplementary Fig. 3). The process of solute segregation during solidification is illustrated by schematics in Supplementary Fig. 2c. The preferential segregation of Ni and Al on the cell boundaries was confirmed by one dimensional (1D) compositional line scans across these boundaries using atom probe tomography (APT) (Fig. 1e). Moreover, due to the constitutional supercooling, the liquid rich in Ni and Al solutes solidifies last as a supersaturated $bcc$ B2 phase on the cell boundaries. The supersaturated prior B2 phase then goes through spinodal decomposition forming an ordered secondary $bcc$-B2 phase and disordered $bcc$-A2 phase of size $\sim10$ nm (Supplementary Fig. 4b)[25]. The APT 1D compositional line scan across the precipitate on the cell boundary confirms the presence of ordered secondary B2 and disordered A2 (Cr-rich $bcc$) phases (Supplementary Fig. 5).

The spinodal mixture of $bcc$ phases, hereafter referred to as (prior) B2, preferentially forms at the cell boundary triple points, because the flux of solute rejection is highest towards these triple

points (Supplementary Fig. 2c), (Fig. 1c). The solidification cells grow in a columnar structure, which forms a honeycomb structure in three dimensions (3D) (Fig. 1a). The equivalent diameter size of the solidification cells is $\sim2.7 \pm 0.4\,\mu m$; the corresponding size of the B2 precipitates on the cell boundaries is $\sim0.54 \pm 0.016\,\mu m$. High-magnification transmission electron microscopy (TEM) images show that these B2 phases on the cell boundaries are connected by nano-bridges of high dislocation density (Fig. 1d). The dislocation nano-bridge form in the regions of the cell boundaries where solutes (Al/Ni) concentration was insufficient to the form B2 phase; however, these solutes trap the dislocations nucleated by thermal cycling during the printing process to form the nano-bridge[26]. The interface of the B2 phase and the $fcc$ phase is incoherent (Supplementary Fig. 4d), indicating that the B2 phase on the cell boundaries can effectively strengthen the microstructure. In contrast, the microstructure of conventionally manufactured (arc-melted) $Al_{0.5}CrCoFeNi$ consists of a dendritic B2 phase of thickness $\sim2$–$30\,\mu m$ (Supplementary Fig. 6), although the size of these B2 dendrites can be reduced by hot rolling[27]. It indicates that the size of the interdendritic B2 phase has been remarkably reduced by a fast-cooling rate during the 3D printing process.

### Strength and ductility
The tensile stress-strain curves of as-printed honeycomb microstructure in Fig. 2a show yield strengths, $\sigma_y$, of $\sim729 \pm 31$ MPa at 298 K and $942 \pm 11$ MPa at 77 K. The elongation to failure, $e_f$, is $\sim16 \pm 4\%$ at 298 K, which increases to $27 \pm 5\%$ at 77 K. Along with the yield strength close to $\sim1$ GPa at 77 K, the representative true stress, $\sigma$, $vs.$ true strain, $\varepsilon$, plots in Fig. 2b indicate that the micro-scale honeycomb microstructure can withstand stresses in excess of 1.5 GPa before fracture at cryogenic temperatures. Furthermore, during tensile loading at 77 K at a true strain exceeding 0.15, the change in slope of the instantaneous work-hardening rate suggests a change in the deformation mechanism. Compared to the properties of its wrought counterpart (grain size $\sim9\,\mu m$)[27] the yield strength of the honeycomb microstructures in the L-PBF $Al_{0.5}CrCoFeNi$ alloy at 298 K is $\sim48\%$ higher. In comparison to other medium- and high-entropy alloys CrCoNi and CrMnFeCoNi, it is respectively $\sim66\%$ and $78\%$ higher at 298 K, and $\sim43\%$ and $24\%$ higher at 77 K[3,5], although its ductility is reduced. The elongation to failure, $e_f$, of the L-PBF alloy is $\sim1.5$–$3$ times lower than that of its wrought counterpart and the CrCoNi-based high/medium-entropy alloys at these temperatures[3,5,27].

### Fracture toughness
The fracture resistance of the honeycomb microstructure was investigated by nonlinear elastic fracture mechanics in the form of $J$-integral-based resistance curves (R-curves) - $J$ $vs.$ crack extension, $\Delta a$ - where $J$ accounts for the contribution of both the elastic and plastic deformation to the nonlinear-elastic strain energy release rate during the fracture process. The $J$-based R-curves for the honeycomb microstructure at 298 K and 77 K are shown in Fig. 2c. At 298 K, the average value of the fracture toughness, $J_{Ic}$, at crack initiation is $\sim406$ kJm$^{-2}$, which marginally drops by $\sim9\%$ to $\sim368$ kJm$^{-2}$ at 77 K. The crack-growth toughness, $J_{ss}$, is determined as the $J$ value at a crack extension of $\Delta a \sim2$ mm allowed for the C(T) specimen geometry in ASTM E1820[28]. The average crack-growth fracture toughness, $J_{ss}$, is $\sim690$ kJm$^{-2}$ and 643 kJm$^{-2}$ at 293 K and 77 K, respectively. All the fracture toughness tests in the present study satisfy the $J$-dominance condition in plane strain, $i.e.$, $B$, $b >> 10(J/\sigma_f)$, where $B$ is sample thickness, $b$ is the width of the uncracked ligament, and $\sigma_f$ is flow stress ($\sigma_f = (\sigma_y + \sigma_u)/2$), where $\sigma_u$ is the ultimate tensile stress. In terms of stress intensity factors, the mode-I fracture toughness, $K_{JIc}$, can be computed through the mode I $J$-$K$ equivalence relationship, $K_{JIc} = \sqrt{J^*(E/(1 - \nu^2))}$, where $E$ is Young's modulus, and $\nu$ is Poisson's ratio determined by ultrasonics method[29]. The average crack-initiation

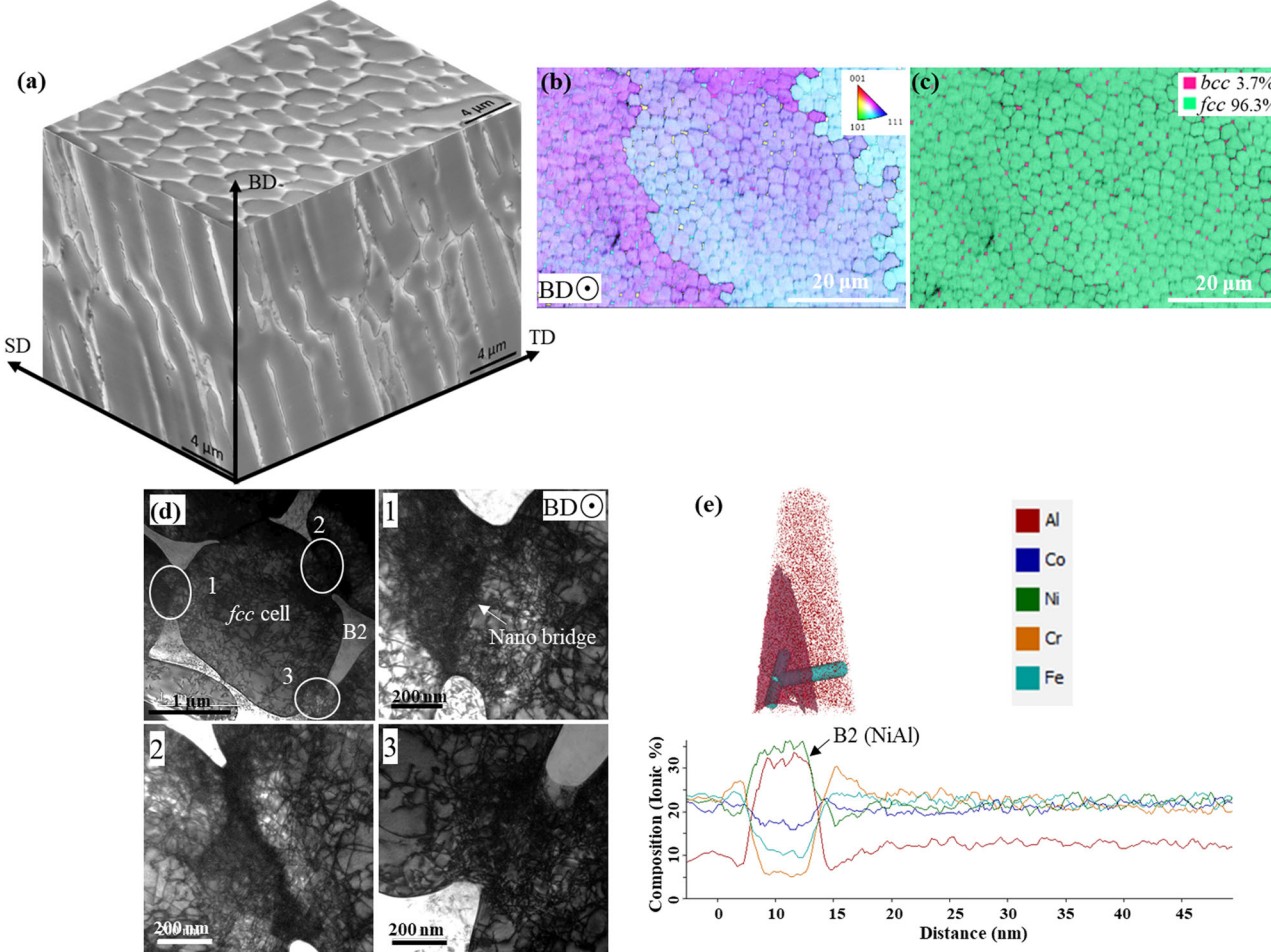

**Fig. 1 | Microstructure of L-PBF as-printed Al$_{0.5}$CrCoFeNi high-entropy alloy. a** A representative pseudo-3D microstructure of solidification cells analogous to a micro-scale honeycomb-like structure. **b** An inverse pole figure (IPF) map showing the near-uniform crystallographic orientation of the solidification cells in different grains and the corresponding phase map in **c** showing the consistent distribution of the B2 (*bcc*) phase on the *fcc* cell boundaries. **d** Bright-field TEM images of a solidification cell show the morphology of the B2 phase on the cellular boundaries, and the magnified images from locations 1, 2, and 3 show the nanometer-scale bridges (nano-bridges) connecting cells, where high-density dislocations are observed. **e** Quantitative compositional analysis by atom probe tomography (APT) shows the distribution of different elements in the *fcc* cells and B2 phase on the cellular boundaries.

fracture toughness, $K_{Jlc}$, at 298 K is ~306 MPa√m which drops marginally by ~5% to 290 MPa√m at 77 K. At these temperatures, the average crack-growth fracture toughness, $K_{Jss}$, are ~399 MPa√m and 385 MPa√m, respectively. Compared to wrought medium-entropy alloy CrCoNi, which is one of the toughest materials on record[4]) and the CrMnFeCoNi Cantor alloy, the fracture toughness, $K_{Jlc}$, of honeycomb microstructure at 298 K is ~47% and 41% higher, respectively. Even at 77 K, the fracture toughness, $K_{Jlc}$, of the present microstructure is ~6% and ~33% higher, respectively[3,5]. The enormity of these crack-initiation toughness, $K_{Jlc}$, results becomes apparent only when it is seen in combination with the yield strength, $\sigma_y$ of the honeycomb microstructure, which is ~66% and 78% higher at 298 K, and ~43% and 24% higher at 77 K compared to the CrCoNi and CrMnFeCoNi alloys [3,5]. In this regard, the current 3D L-PBF-printed Al$_{0.5}$CrCoFeNi alloy has superior damage-tolerance, in terms of a combination of strength *and* toughness, than the well-known wrought CrCoNi and CrMnFeCoNi medium- and high-entropy alloys.

To further elucidate the uniqueness of the microstructural design proposed here and its role in the intrinsic toughening of Al$_{0.5}$CrCoFeNi, a set of four heat-treatments (Supplementary Fig. 7) were performed to achieve the optimal combination of yield strength, $\sigma_y$, strain hardening, and elongation to failure, $e_f$. The heat treatments did not affect the morphology of the B2 precipitates on the cell boundaries; however, it led to the formation of needle shaped B2 precipitates of various dimensions in the supersaturated *fcc* matrix, i.e., the interior of the cells (Supplementary Figs. 8 and 9). After an annealing treatment at 750 °C for one hour (HT3), the microstructure consists of uniformly distributed B2 needles of multiple dimensions (Supplementary Figs. 9d, 10d). Out of the four heat treatments, the HT3 microstructure shows the best combination of tensile properties including strain hardening, *i.e.*, $\sigma_y$ ~1090 MPa and $e_f$ ~9% at 298 K, $\sigma_y$ ~1376 MPa and $e_f$ ~6.6% at 77 K (Supplementary Fig. 11). Therefore, C(T) specimens were heat-treated following the HT3 schedule for fracture toughness tests. The *J*-integral, *J*, *vs.* crack extension, $\Delta a$, results after the annealing treatment are plotted in Fig. 2d and with enlarged view in Supplementary Fig. 12. At 298 K, the HT3 microstructure shows a rising R-curve behavior with crack-initiation fracture toughness, $K_{Jlc}$, of ~135 MP√m, and crack-growth fracture toughness, $K_{Jss}$, of ~168 MPa√m, which is still reasonable considering the $\sigma_y$ ~1090 MPa yield strength. However, at 77 K, the crack grows unstably, resulting in an initiation fracture toughness of $K_{Jlc}$ ~ 51 MPa√m. The crack-initiation fracture toughness, $K_{Jlc}$, of the annealed microstructure is less than half at 298 K and ~5.7 times lower at 77 K compared to the honeycomb microstructure. During tensile and fracture toughness tests, the honeycomb microstructure fractured by microvoid nucleation and coalescence at both 298 K and 77 K temperatures

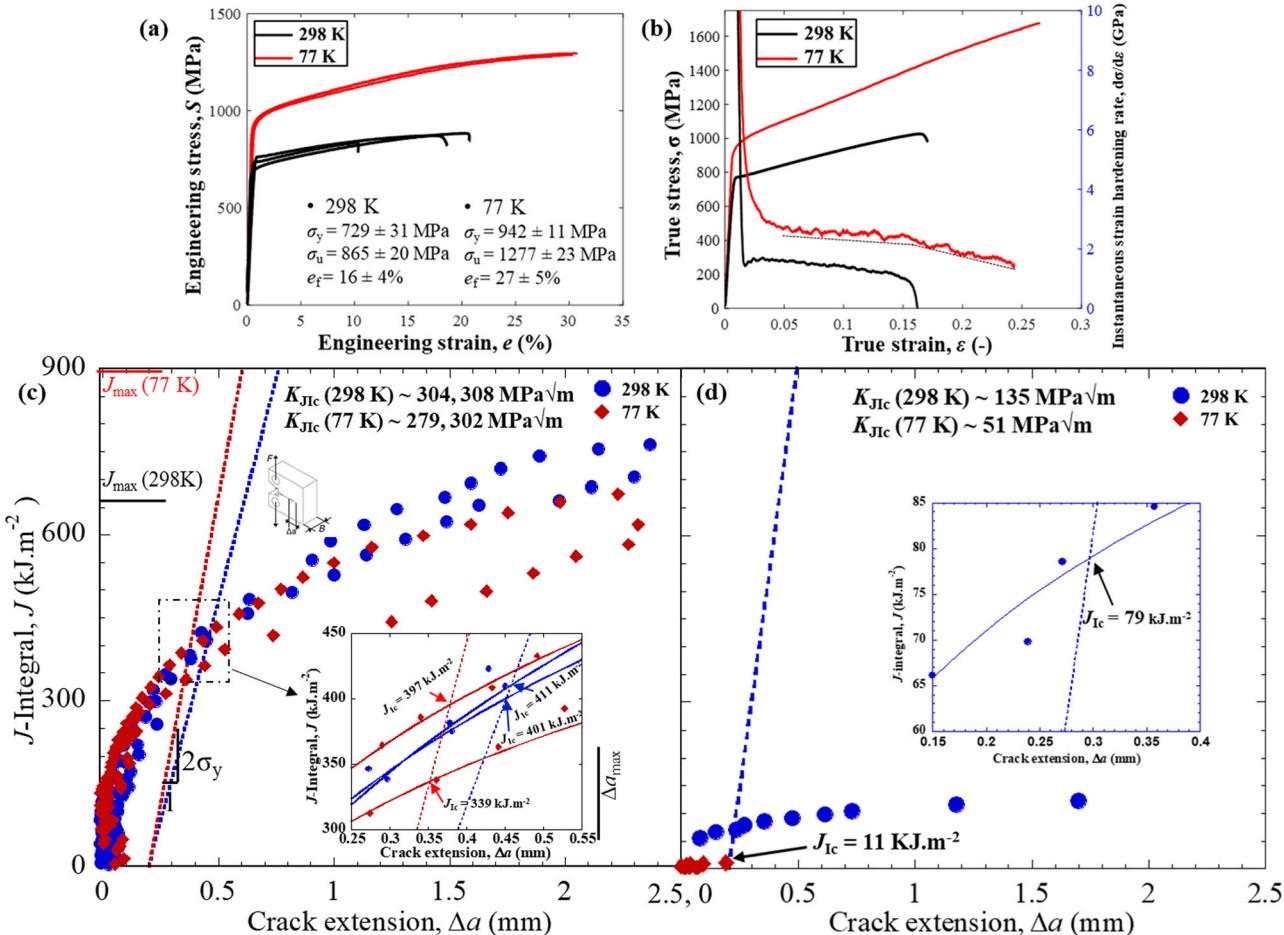

**Fig. 2 | Mechanical properties of L-PBF printed Al$_{0.5}$CrCoFeNi high-entropy alloy at 298 K and 77 K. a** Tensile curves at 298 K and 77 K show yield strength, $\sigma_y$, and ultimate tensile strength, $\sigma_u$ increasing by ~29% and 48%, respectively, at cryogenic temperatures. **b** Representative true stress, $\sigma$, and instantaneous strain hardening rate, d$\sigma$/d$\varepsilon$ *vs.* true strain, $\varepsilon$ plot for specimens tested at 77 K show a change in the slope of the work hardening rate with increasing true strain, $\varepsilon$. The change in slope at the cryogenic temperature indicates a varied deformation mechanism. **c** The crack-resistance curves (R-curves) show average fracture toughness, $K_{JIc}$ of ~306 MPa√m and ~291 MPa√m at 298 K and 77 K, respectively. **d** The fracture toughness, $K_{JIc}$ of the specimen after the heat treatment at 750 °C for 1 h drops to ~135 MPa√m and ~51 MPa√m at 298 K and 77 K, respectively.

(Supplementary Figs. 13, 14). However, the HT3 microstructure also fractured by microvoid coalescence at 298 K (although the dimples were relatively shallow); at 77 K the HT3 microstructure displayed a brittle fracture where it cleaved along the cell boundaries (Supplementary Figs. 15, 16).

### Deformation mechanisms

To understand the reason behind the excellent combination of yield strength, $\sigma_y$, and fracture toughness, $K_{JIc}$, of the honeycomb microstructure, the post-fracture specimens were sectioned from mid-thickness to investigate the deformation mechanisms in the fully plane-strain region. An electron backscattered diffraction (EBSD) inverse pole figure map in Fig. 3a shows the crack propagation path and deformed microstructure around it in a sample fractured at 77 K. The arrows in Figs. 3a, b indicate misorientation bands emanating from the crack. A correlated weighted Burgers vector (WBV) map in Supplementary Fig. 17, corresponding to the EBSD map in Fig. 3a, reveals the distribution of dislocations around the crack tip. WBV maps in Supplementary Figs. 17 and 18 illustrate that the misorientation bands emanating from the crack tip correspond to a band of dislocations moving away from the crack tip; indeed, the dislocation bands spread as far as ~300 μm away from the tip. A comparison of WBV maps before (Supplementary Fig. 1d) and after deformation indicate that these bands of dislocations form during crack initiation and propagation. Moreover, the magnified EBSD IPF and WBV maps in Figs. 3b, c show that they pass through the nano-bridges/gaps available in between the B2 phase precipitates on the triple points of the cellular boundaries. A montage of the TEM images in Fig. 3d, from the specimen tested at 298 K, also illustrates that dislocations move through the gaps between the B2 phase (nano-bridges) on the cell boundary. Furthermore, at 77 K, the TEM image from the plastic zone ahead of the crack tip shows deformation-induced nano-twinning (Fig. 3e). The growth of these nano-twins is restricted by the B2 phase on the *fcc* cell boundaries (Fig. 3f and Supplementary Fig. 19), which also contributes to the toughness of the microstructure. The change in the slope of the work-hardening rate during tensile deformation at 77 K also indicates activation of twinning above a true strain of $\varepsilon > 0.15$ (Fig. 2b). Post-fracture deformation mechanism near the crack tip in the HT3 microstructure was also investigated. Supplementary Fig. 20 shows the EBSD IPF and WBV maps of HT3 microstructure fractured at 298 K and 77 K, respectively. At 298 K, the WBV map shows the deformed region with high dislocation density spreading over a distance of ~50 μm from the crack tip; however, at 77 K, the extent of such plastic deformation is restricted to a maximum distance of ~10 μm.

### Discussion

The microstructure of LPBF Al$_{0.5}$CrCoFeNi is different than the cellular structure usually present in additively manufactured alloys. Usually,

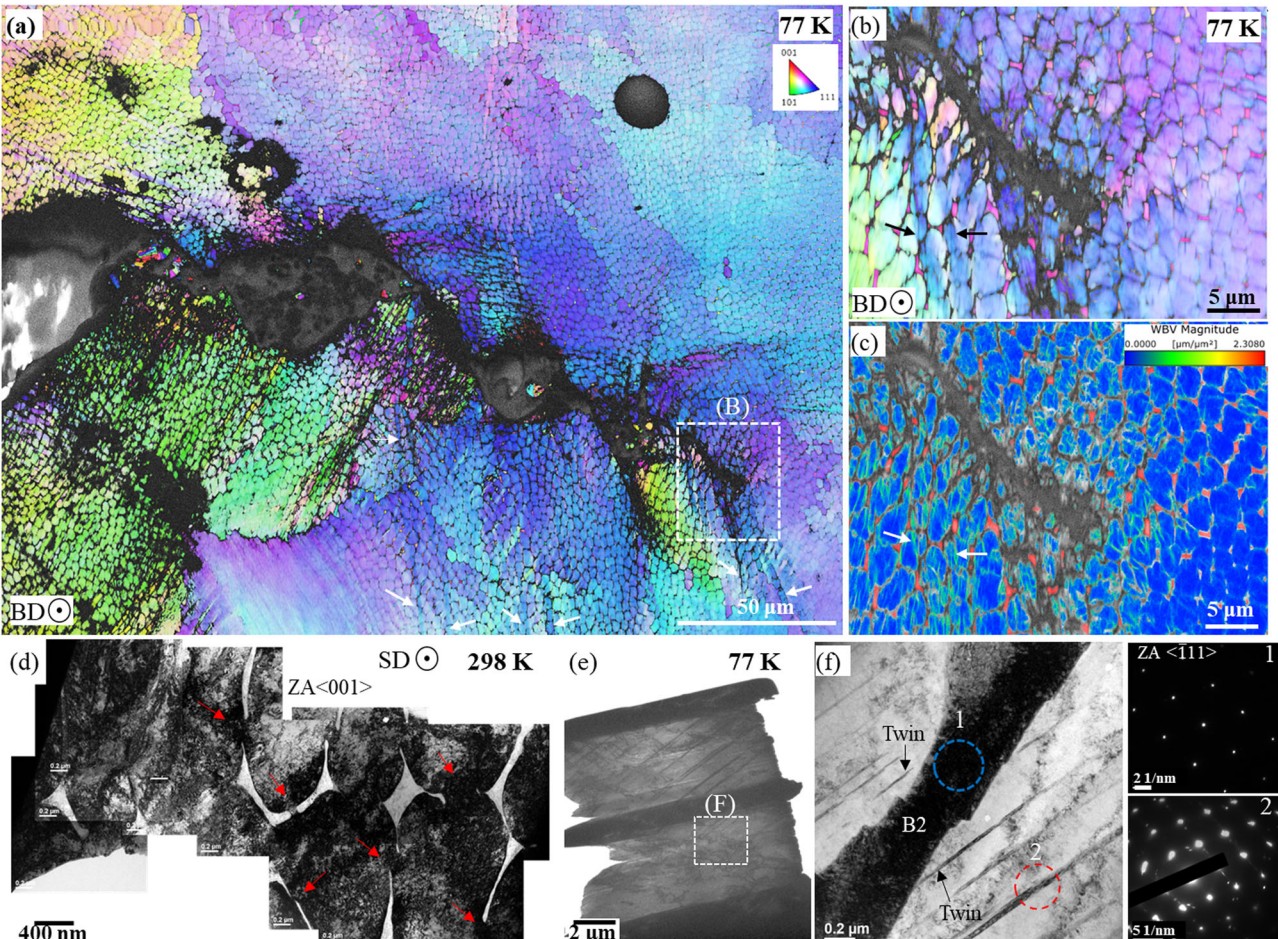

**Fig. 3 | Deformation mechanism in L-PBF as-printed Al$_{0.5}$CrCoFeNi high-entropy alloy at 298 K and 77 K.** Compact-tension C(T) specimens tested at 298 K and 77 K were sectioned from the mid-plane to expose the crack-tip region in the plane-strain condition; this region near the crack tip was examined by EBSD and TEM. **a** The EBSD IPF map from the wake of the crack in the specimen tested at 77 K shows bands of misorientation spreading away from the crack (indicated by white arrows). **b** A magnified IPF map near the crack tip shows these misorientation bands traverse through the honeycomb cellular structure (indicated by black arrows), as well as the deflected cracks along cellular boundaries. **c** The weighted Burgers vector map shows that the misorientation bands are highways for dislocation movement allowing the plasticity to spread further from the crack tip. **d** TEM near the crack-tip region of the specimen tested at 298 K shows that the B2 phase on the cell boundaries blocks the shearing of dislocations; meanwhile, the nano-bridges allow the transit of dislocations (indicated by the red arrows), thereby simultaneously strengthening and toughening the material. **e** TEM image from near the crack-tip region of the specimen tested at 77 K shows nano-twins formed during the cryogenic deformation. **f** the nano-twins are blocked by the B2 phase on the cell boundary. Diffraction patterns from location 1, and 2 confirms the presence of the B2 phase and the nano-twins, respectively.

the cellular boundaries are decorated by solute atoms and intertwined dislocations. However, in LPBF Al$_{0.5}$CrCoFeNi, the cellular structure consists of a secondary phase B2 with interconnecting dislocation nano-bridges. The B2 phase on the cell boundaries restricts the movement of dislocations and the nano-twins during deformation at 298 K and 77 K (Supplementary Fig. 19). The strenuous movement of dislocations and restriction to the growth of deformation nano-twins promote strain delocalization and necessitate additional energy to open the crack tip. However, to avoid crack extension by brittle fracture, it is also important to spread the damage away from the crack tip[3,30]. Thereby, in case of severe plastic deformation, i.e., close to the crack tip, the nano-bridges in between the B2 phase breaks down to facilitate the movement of dislocations away from the crack tip (indicated by white arrows in Fig. 3a). As evident from Figs. 3a, b, the strain localization in the ductile *fcc* cell is insufficient to initiate debonding of the hard *bcc* phase present on the cell boundaries. In combination, the nano-bridged honeycomb microstructure of the LPBF Al$_{0.5}$CrCoFeNi provides a unique deformation mechanism resulting in a strength (~1 GPa) and toughness (~300 MPa√m). However, in the case of the LPBF CrCoNi with a conventional cellular structure, the combination of

strength and toughness is even inferior to their conventionally manufactured counterpart[31] (Fig. 1), which highlights the uniqueness of the cellular structure in LPBF Al$_{0.5}$CrCoFeNi.

The "banana plot" in Fig. 4 illustrates that the honeycomb nano-bridge microstructure of the L-PBF Al$_{0.5}$CrCoFeNi shows one of the best combinations of yield strength, $\sigma_y$, and fracture toughness, $K_{Jlc}$, compared to all known structural materials. The multiphase honeycomb microstructure is tougher and stronger than recently developed conventional single-phase high- and medium-entropy alloys and the high Mn/Ni cryogenic steels[5–7,32,33]. The honeycomb microstructure (in the as-printed condition) consists of a high dislocation density inside the cells (Fig. 1d) with the B2 phase on the cell boundaries connected by nano-bridges of dislocations[26,34,35] (Fig. 3d). Both, in combination with solid solution strengthening, provide a yield strength, $\sigma_y$, ~48% higher compared to the wrought counterpart at 298 K[27]. Furthermore, during severe plastic deformation, dislocations can move through the nano-bridges connecting the B2 precipitates on the cell boundaries (Fig. 1d). Therefore, the microstructure shows a crack-initiation fracture toughness, $K_{Jlc}$, ~306 MPa√m, by spreading the plastic damage further from the crack tip. Moreover, by activating nano-twinning at

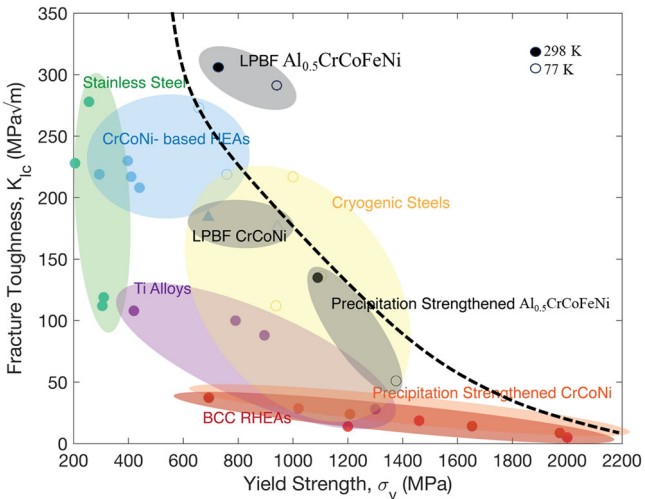

**Fig. 4 | A "banana plot" of fracture toughness as a function of the yield strength of various classes of materials**[3–5]**.** The nano-bridged honeycomb cellular structure formed by L-PBF 3D printing is stronger and tougher than both cryogenic steels and the recently developed *fcc* class of - and medium-entropy alloys produced by conventional methods. This plot showcases the excellent combination of strength and fracture toughness of the additively manufactured Al$_{0.5}$CrCoFeNi.

77 K in combination with dislocation-assisted plastic deformation, the honeycomb microstructure shows a yield strength, $\sigma_y$, ~1 GPa, and fracture toughness, $K_{JIc}$, ~300 MPa√m. It should be noted that by manipulating the process parameters of the 3D printing, i.e., the thermal gradient, $G$, and the solidification rate, $R$, the size of the honeycomb cells can be altered (Supplementary Fig. 2)[17]. Consequently, the material's strength and fracture toughness can be tailored to suit particular application requirements. The multiphase honeycomb microstructure produced by L-PBF, a "bottom-up" manufacturing process, highlights the possibility of alloy design for intrinsic toughening. In contrast, the uniformly distributed B2 precipitates after heat treatment completely restrict the movement of dislocation away from the crack tip (<10 μm) resulting in ~5.7 times less fracture toughness at 77 K. These two extreme results highlight the limitation of conventional "top-down" processes in simultaneously strengthening and toughening ductile materials.

Nature builds structures "bottom-up" to support different structural requirements at multiple length-scales in the natural materials[2,11]. The honeycomb microstructure of Al$_{0.5}$CrCoFeNi produced by "bottom-up" L-PBF 3D printing demonstrates the possibility of strengthening in combination with the intrinsic toughening of ductile materials. Here, the strengthening is driven by restricting the dislocation motion before the start of uniform plastic deformation; however, during severe plastic deformation, the dislocation movement away from the crack tip is facilitated to improve the fracture resistance through blunting and shielding of the crack tip.

Face-centered cubic medium- and high-entropy alloys, particularly involved CrCoNi-based alloys have shown excellent mechanical properties when conventionally ("top-down") processed;[3–5] indeed, the equiatomic CrCoNi alloy, which displays a tensile strength exceeding 1 GPa, is one of the toughest reported materials on record[4] (although the initial yield strength can be below 1 GPa). These alloys have a remarkably simple, single-phase microstructure of nominally equiaxed grains, yet derive their strength and toughness from a synergistic sequence of deformation mechanisms, involving dislocation slip, stacking-fault formation, nano-twinning and phase transformation, to generate prolonged and continuous strain hardening; the strain hardening increases strength yet at the same time delays plastic

instability by necking to promote ductility – the "perfect storm" to create very high toughness[4].

Here we show that the damage-tolerance of face-centered cubic medium-and high entropy alloys, i.e., the desired combination of yield strength and fracture toughness, can actually be manipulated by using additive ('bottom-up') manufacturing, in the present study by laser powder bed fusion that can generate additional internal structure in the form of hierarchical honeycomb microstructures, to further enhance their strength properties without compromising their fracture toughness. The alloy is strengthened by restricting the movement of dislocations at the early stage of plastic deformation; however, during severe plastic deformation, i.e., near the crack tip, the dislocation movement is facilitated by the breaking down of dislocation nano-bridge connecting the B2 phase, which improves the fracture resistance through blunting and shielding of the crack tip.

## Methods
### Materials processing
A pre-alloyed powder of composition (wt.%) Cr ~22.5%, Co ~24.6%, Fe ~23.2%, Ni ~24.5%, with Al ~5.21% was used to fabricate the Al$_{0.5}$CrCoFeNi blocks by the laser powder bed fusion (L-PBF) process. The gas-atomized pre-alloyed powders were supplied by Vilory Advanced Materials Technology with a powder size distribution of ~25–63 μm. The specimens were printed using an SLM280 (SLM Solutions, Germany) machine with an Nd: YAG fiber laser. The printing process was performed under a protective Ar gas environment with an optimized set of processing parameters laser power ~300 W, layer thickness ~50 μm, hatch spacing ~90 μm, and a scanning speed ~500 mm.s$^{-1}$ with a relative density of ~99.9%. The orientation of the 3D printed block with respect to the build platform is shown in Supplementary Fig. 21a. The bidirectional stripe scanning strategy employed for the process is illustrated by the schematic in Supplementary Fig. 21b.

Four different heat treatments at temperatures 650 °C (20 h), 700 °C (1 and 2 h), and 750 °C (1 h) were performed on as-printed samples following the steps shown by schematics in Supplementary Fig. 7. The samples were heat-treated in a box furnace and followed by water quenching. Blocks of similar thickness were heat-treated for fracture toughness and tensile properties evaluations.

### Microstructural characterization
The composition of the powder and the bulk 3D printed specimens was measured by inductively coupled plasma atomic emission spectroscopy (ICP-AES), combustion infrared detections, and inert gas fusion were used to measure the interstitials C, O, S, and N. The microstructure of the additively manufactured specimens was investigated using optical and electron microscopes after mechanically polishing them to a mirror finish. The spatial composition in the microstructure was measured by one dimensional line scans using atomic probe tomography (Supplementary Figs. 5, 23). Some polished specimens were etched using Kalling's reagent to observe the melt pool boundaries and laser scan tracks. Electron back-scattered diffraction (EBSD) mapping with a step size of ~100 nm was used to analyze the crystallographic texture and distribution of *fcc* and B2 phases in the microstructure. The dislocation density in the specimens before and after the deformation was mapped, measuring every point's weighted Burgers vector (WBV). A commercially available software, AZtecCrystal (Oxford Instruments), was used for mapping the WBV from the EBSD scans. The WBV is the sum of all types of dislocations defined as ~the density of intersection of dislocations with a map × Burgers vector[36]. The Burgers circuit in this method is drawn on the sample frame of reference[37]. During the analysis, the elastic strain is assumed to be small; therefore, the lattice distortion is entirely from the dislocations. There are no assumptions regarding the

orientation gradient in the third dimension; therefore, the dislocation density measured by this method is relatively more accurate than other 2D mapping methods[38]. The WBV method is also suitable for dislocation density mapping in microstructure with cellular structure. The magnitude of WBV, when defined as a coordinate invariant, gives the lower bound of the magnitude of dislocation density tensor[36]. A high-resolution transmission electron microscope (HRTEM) was used to analyze the solidification cellular structure with the B2 phase on their boundaries. For HRTEM, the specimens were sectioned into disks of diameter ~3 mm and then mechanically ground to a thickness ~100 μm. Further, these disks were thinned by twin-jet polishing at −20 °C under a stable current of ~25 mA. The high-resolution quantitative chemical analysis of the 3D-printed HEA samples was performed using the atom probe tomography (APT) technique (CAMECA, LEAP 3000X HR). The lift-out and annular milling technique was used to prepare micro-tip specimens of an apex radius of ~ 50 nm site, specifically with the FEI Helios dual-beam focused ion beam (FIB). The APT micro-tip specimens were analyzed at 40 K with pulses of green laser light (532 nm wavelength) at a 200 kHz repetition rate, an energy of 0.9 nJ pulse$^{-1}$ and an evaporation rate of 0.30%. Data analysis was performed using IVAS 3.8.4 software[25].

For analyzing the microstructure after deformation, the C(T) specimens were sectioned at mid-thickness to expose the region fully under plane strain during the fracture toughness tests (Supplementary Fig. 22). One-half of the C(T) specimens was analyzed by EBSD to observe the crack path and the nearby microstructure. The other half of the specimens were used for TEM to investigate the deformation mechanism near the crack tip. For this, the TEM specimens were sectioned by focus ion beam (FIB) from inside the plastic zone, $r_p$ (~1/$2\pi(K_{JIc}/\sigma_y)^2$) (Supplementary Fig. 22). A JEOL 2010 TEM under an accelerating voltage of 200 kV was used to characterize the deformation mechanism near the crack tip. The fracture surfaces of C(T) and tensile specimens were analyzed using secondary electron detectors in a scanning electron microscope.

## Mechanical properties characterization

The specimens for mechanical tests at 298 K and 77 K were machined by wire electrical discharge machining (EDM) from the as-printed blocks in their respective orientations, as illustrated by the schematic in Supplementary Fig. 21a. The dog bone tensile specimens of the gauge length, $L$ ~16.8 mm, were used for the tensile tests (Supplementary Fig. 21d). Before the tests, the tensile specimens were mechanically polished to remove the oxide layer formed on their surface during the wire EDM process. The tensile tests were conducted on a servo-hydraulic load frame (MTS Corp., Eden Prairie, MN, USA) operated by an Instron digital controller (Instron Corp., Norwood, MA, USA) at a strain rate of 10$^{-3}$ s$^{-1}$ as recommended in ASTM Standard E8[39]. An Epsilon clip-on extensometer suitable for use in the temperature range of 4 K to 498 K was attached to the specimen to measure the elongation during tensile loading.

For the fracture toughness tests, C(T) specimens of width, $W$ ~18 mm, and thickness, $B$ ~11 mm were machined from the as-printed and heat-treated blocks with the direction of V-notch perpendicular to the build direction (Supplementary Figs. 21a, c). The notch root radius, ρ (~150 μm) of the V-notch sectioned by the wire EDM was further reduced to ~20–50 μm by a mechanical micro-notch machine equipped with a lubricated razor blade. A relatively sharp notch in specimens promotes a uniform crack initiation through the thickness during fatigue precracking, and it also reduces the initial load required to initiate the fatigue crack. Before precracking, the C(T) specimens were polished to a mirror finish to monitor the crack growth from the surface. The crack length during precracking was measured using a long-distance microscope equipped with a digital camera. The C(T) specimens were fatigue precracked on a 100 kN servo-hydraulic 810 MTS load frame (MTS Corp., Eden Prairie, MN, USA) operated using an

Instron 8800 digital controller (Instron Corp., Norwood, MA, USA). For precracking, a sinusoidal waveform of constant amplitude, load ratio ~0.1, and frequency of 15 Hz were used. The total crack length of $a/W = 0.45 – 0.55$ was achieved by precracking, where $a$ is the crack length, and $W$ is the width of the specimen. The precracked specimens were side grooved by ~1 mm on each side to ensure a straight crack path during the fracture toughness tests. The nonlinear-elastic fracture toughness measurements were performed to measure both the elastic and in-elastic contribution to the fracture resistance. Based on the methodology proscribed in ASTM E1820[28] the R-curve behavior of the materials, i.e., $J$-integral as a function of crack extension ($\Delta a$), was measured. The fracture toughness tests were conducted at a cross-head velocity of 0.02 mm.s$^{-1}$, with the crack extension during the tests determined using a compliance method from the measurements of load-line displacement. A clip gauge of 3 mm gauge length (Epsilon Technology, Jackson, WY, USA), capable of measuring displacement in the temperature range of 4 K to 498 K, was used to measure the load-line displacement. The crack length, $a_i$, was obtained using the following equation based on rotation-corrected unloading compliance:[28]

$$ai/W = 1.000196 - 4.06319u + 11.242u^2 \\ -106.043u^3 + 464.335u^4 - 650.677u^5, \tag{1}$$

where

$$u = \frac{1}{\left[B_e E C_{c(i)}\right]^{1/2} + 1}, \tag{2}$$

where $B_e$ is the effective thickness of the side-grooved sample defined as $B_e = B \cdot (B-B_N)^2/B$, $B_N$ is the thickness of the specimen at the side groove, $C_{c(i)}$ is the elastic unloading compliance after correction for the rotation during the crack-tip opening, and $E$ is the material's elastic modulus. The final crack length was also verified from the fractograph of the fractured samples (Supplementary Fig. 14). To distinguish the region of the crack growth during the fracture toughness tests, the specimens were fatigued to failure after the fracture toughness tests.

The sum of $J_{el(i)}$ and $J_{pl(i)}$ for the corresponding crack length, $a_i$, gives the total measurement of $J_i$-integral, i.e., $J_i = K_i^2/E' + J_{pl(i)}$, where $E' = E/(1 - v^2)$, v is Poisson's ratio, and $K_i$ is the linear stress intensity factor at the crack tip corresponding to the load-displacement curve. For the geometry of C(T) specimens, the $K_i$ can be determined from the equation:

$$Ki = \frac{P_i}{(BB_N W)^{1/2}} f(a_i/W), \tag{3}$$

where $P_i$ is the applied load for every corresponding point and $f(a_i/W)$ is a geometry-dependent function as listed in ASTM E399[40]. The plastic component, $J_{pl(i)}$, can be computed using the following equation:

$$Jpl(i) = \left[J_{pl(i-1)} + \left(\frac{\eta_{pl(i-1)}}{b_{(i-1)}}\right)\frac{A_{pl(i)} - A_{pl(i-1)}}{B_N}\right]\left[1 - \gamma_{(i-1)}\left(\frac{a_{(i)} - a_{(i-1)}}{b_{(i-1)}}\right)\right], \tag{4}$$

where $A_{pl(i)}$-$A_{pl(i-1)}$ is incremental plastic area under load-displacement curve, $\eta_{pl(i-1)} = 2 + 0.522 \, b_{(i-1)}/W$, and $\gamma_{pl(i-1)} = 1 + 0.76 \, b_{(i-1)}/W$. Here, $b_i$ is the uncracked ligament width, i.e., $b_i = (W-a_i)$. $J_i$ can be determined for the corresponding crack extension using the above equation. From the $J$-$\Delta a$ curve (R-curve), where $\Delta a = a_i$-$a_o$ and $a_o$ is crack length after fatigue precracking, the provisional fracture toughness, $J_Q$, can be determined from 0.2 mm offset/blunting line ($J = 2\sigma_f \Delta a$, where $\sigma_f = \frac{\sigma_y + \sigma_u}{2}$). These $J_Q$ measurements satisfied the $J$-dominance and plane-strain conditions for validity, i.e., $b_o$, $B > 10 \, J/\sigma_f$, where $b_o$ is the uncracked ligament length ($W$-$a$), $B$ is the specimen thickness;

accordingly, the measured critical $J$ value at crack initiation can be defined as size-independent fracture toughness $J_{Ic}$. From this measured $J_{Ic}$, the equivalent Mode I fracture toughness $K_{JIc}$ can be computed using equations: $K_{JIc} = (J_{Ic}*E)^{0.5}$ and $K_{Jss} = (J_{SS}*E)^{0.5}$. Here $J_{ss}$ is steady-state fracture toughness defined as the maximum valid measurement of $J$ value based on geometry considerations of the C(T) specimens in corresponding environmental conditions; this is identified as the crack-growth fracture toughness. The elastic modulus $E$ of the specimens and Poisson's ratio, $\nu$, were measured using the non-destructive ultrasound spectroscopy method[3]. The measured values of Young's modulus, $E \sim 218 \pm 3.4$ GPa, and Poisson's ratio, $\nu \sim 0.23$, were used for computing $K_{JIc}$ at 298 K and 77 K assuming a minimal change in $E$ at these temperatures as seen in the case of alloys of similar compositions[4,5]. All six fracture toughness measurements in the present study satisfied the criteria for $J$-dominance and plane-strain conditions.

## Data availability

All data that support the findings of this study are available within the Article and the Supplementary Information. Further information can be obtained from the corresponding authors.

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

## Acknowledgements

This work was primarily supported by the U.S. Department of Energy, Office of Science, Office of Basic Energy Sciences, Materials Sciences and Engineering Division under contract no. DE-AC02-05-CH11231 to the Damage-Tolerance in Structural Materials Program (KC13) at the Lawrence Berkeley National Laboratory (LBNL). S.H. and U.R. acknowledge the support of the Structural Metal Alloys Program (Grant reference no.:A18B1b0061) of the Agency for Science, Technology and Research of Singapore. D.H.C. acknowledges support of an NSF Graduate Fellowship from the National Science Foundation under Grant No. DGE 2146752. The authors appreciate the help of Michael Gronley from the machining shop in the Engineering Division of LBNL. X.T. acknowledges the support of NUS Start-up Fund (22-5928-A0001) and Singapore Ministry of Education Academic Research Fund Tier 1 grant (22-4902-A0001). The authors would like to acknowledge the help received from H. Zhang, C. Zhang, B. Dong, A. Patnaik, and L.N. Ramasubramanian. for the focus ion beam, electron microscopy, and heat treatment experiments, and D. Mangelinck. and METSA network for the APT experiments.

## Author contributions

P.K., X.T., and R.O.R. conceived the idea and led the project. S.H., X.T., and U.R. helped with the 3D printing of the specimens. P.K., D.C., and S.H., conducted the mechanical testing and microstructural characterization by SEM/EBSD, K.C. and X.T. helped with TEM analysis, X.T. helped with APT experiments, P.K. wrote the manuscript, X.T., U.R., and R.O.R. helped with editing.

## Competing interests

The authors declare no competing interests.
