## [Peer Review File · Nature Communications]

A strong fracture-resistant high-entropy alloy with nano-bridged honeycomb microstructure: Intrinsically toughened by 3D-printingREVIEWER COMMENTS

Reviewer #1 (Remarks to the Author):

In this research, the honeycomb microstructure of $\text{Al}_{0.5}\text{CrCoFeNi}$, produced by "bottom-up" L-PBF 3D printing, demonstrates the potential for enhanced strength along with the intrinsic toughening of ductile materials. This cellular structure features a secondary phase interconnected with dislocation nano-bridges, offering a unique deformation mechanism that results in an exceptional combination of strength (~ 1 GPa) and toughness (~ 300 MPa $\sqrt{\text{m}}$). Despite recognizing the significance of this superior properties, a more detailed explanation of the mechanism behind the high toughness is required. The following questions arise.

Nano-bridges form during solidification due to the formation of the B2 phase at the cell boundary. During severe plastic deformation, dislocations can traverse these nano-bridges, connecting the B2 precipitates on the cell boundaries. If we consider functionality solely in terms of the dislocation path, thicker bridges between two cells might be advantageous. So, why should our focus be on the nano-bridge?

For fabrication, HT3 was used on the C(T) specimen to evaluate fracture toughness. How did microstructural evolution take place during heat treatment at 750 °C? How does this differ from the heat treatment carried out at lower temperatures (650 and 700 °C) over an extended period? The microstructure comprises B2 precipitates of various sizes, uniformly distributed. How were the nanoscale B2 precipitates uniformly distributed in the microstructure after the heat treatment (HT3)? Why were more nano-bridges formed during HT3? A thermodynamic explanation is needed.

At 298 K, the strength of HT4 is the highest, but at 77 K, that of HT3 is the highest. How do the microstructures of each differ? Why does HT3 have the highest strength at 77 K? If it's attributed to the formation of nano-bridges, please provide a quantified analysis for that.

In the supplementary materials, Fig. 7d, two types of B2 precipitates are present: one is the whisker-type nano-precipitates, and the other is the rounded (sometimes elliptical) micro-precipitates. How do the roles of each phase differ?

It seems that the B2 phase negatively impacts mechanical properties. However, without the B2 phase, nano-bridges cannot exist in the same way. Thus, the role of the B2 phase should be detailed. In Fig. 3, the deformation-induced nano-twins are obstructed by the B2 phase at the cell boundary. What role does the B2 phase play in fracture toughness? What is the optimal volume fraction, size, distribution, and morphology of the B2 phase to enhance both yield strength and fracture toughness?

In the Supplementary Materials Fig. 20, APT analysis was conducted on the fcc cell, intercellular B2, and the BCC phases. Nano-twins and dislocations are primarily obstructed by the B2 phase but traverse through the nano-twins, which is advantageous for enhancing toughness. The high dislocation density is thoroughly discussed in numerous figures, but I'm intrigued by the APT results at the boundary between the nano-bridge and the B2 phase. If the authors can clarify dislocation behavior in terms of atomic distributions, it would bolster the paper's argument by elucidating the channeling effect of nano-bridges.

The following are minor comments:

Even though the authors reference ref. 25, the explanation for Supplementary Materials Figure 3 is insufficient. Please indicate the zone axis in the TEM DP and provide a detailed explanation of the ordered B2 and disordered A2 phase.

Line 436-439:

In refs. 4 and 5, please remove "(1979)".

Line 211:

Please change "Figure" to "Fig."

Supplementary Materials Figures 18, 19, and 20 lack explanations in the manuscript.

Reviewer #2 (Remarks to the Author):

The authors present a novel and interesting study, wherein the Al_{0.5}CrCoFeNi high-entropy alloy (HEA) develops a unique microstructure due to its fabrication via laser-beam powder bed fusion process. Many interesting aspects regarding the formation of BCC-FCC microstructure and the corresponding enhancement in strength-ductility/toughness combination are explained in depth. It is well written, and thus suggested to be accepted after the following comments are addressed:

(1) The distinction between the A2 and B2 phases is important in the Al_{0.5}CrCoFeNi HEA system and shall be elaborated further. How do the authors clearly distinguish between the ordered and disordered BCC phases via the TEM investigations (supplementary materials Figure 3) and EBSD (Fig. 1c)?

(2) The authors mention "honeycomb" (in several instances, for example, Line 115) microstructure in reference to the cell boundaries of the as-printed alloy. The cellular structures only have a typical irregular hexagonal shape (M. Moneghan, C. Williams, R. Mirzaeifar, J. Mater. Res. 2020, 35, p.1984) and the presence of strengthening advantages from the naturally occurring "honeycomb" structure is debatable in the 3D-printed HEA.

(3) The as-built microstructure has been termed as "multiphase" (Line 244). Unless the A2 and B2 phases are clearly demarcated and their role in deformation is explicated, the multiphase nature of the HEA may be questionable.

(4) The 3D-printed metallic structures are well known to be anisotropic, wherein the large columnar grains grow along the building direction. However, it is interesting to note that even the cellular structure (which is generally randomly oriented, Y.-K. Kim, S. Yang, K.-A. Lee, Addit. Manuf. 2020, 36, 101543) in the current study (Fig.1a) seems to be anisotropic and elongated along the building direction. The related explanations on this will add significant value to the microstructural discussion.

(5) The building and scanning directions shall be indicated in Figure 3.

(6) What is the equilibrium partition coefficient of Ni and Al, and how does it compare with those of Cr, Co, and Ni, such that the segregation of Ni and Al took place in the cell boundaries? Can the authors indicate what factors influence such segregation and the equilibrium partition coefficient during laser beam powder bed fusion?

(7) It is difficult to state whether the crack growth is chiefly along the cell boundaries or through the cell interiors. The cell interiors consist of a more ductile FCC phase whereas the harder BCC phase is present on the cell boundaries. The deformation mechanism explained on page 8 (Lines 197-208) could be elaborated further in this respect.

(8) Supplementary Materials Figure 18. Why the data points first move towards left (i.e., the crack extension ($\Delta(a)$) decreases or the crack becomes shortened) with increasing J-integral value needs to be explained.

Responses to the Reviewers' comments

At the outset, we thank the reviewers for carefully reading our paper and providing their detailed comments. We respond to the comments below (our responses are in blue.) We also highlight the changes made to the text in red wherever necessary.

Reviewer #1:

General comment: Reviewer #1: In this research, the honeycomb microstructure of Al_{0.5}CrCoFeNi, produced by "bottom-up" L-PBF 3D printing, demonstrates the potential for enhanced strength along with the intrinsic toughening of ductile materials. This cellular structure features a secondary phase interconnected with dislocation nano-bridges, offering a unique deformation mechanism that results in an exceptional combination of strength (~1 GPa) and toughness (~300 MPa√m). Despite recognizing the significance of this superior properties, a more detailed explanation of the mechanism behind the high toughness is required. The following questions arise.

Response: We thank the Reviewer for the encouraging comment. We have revised the manuscript to clarify the mechanism behind the high fracture toughness. We address their specific comments below.

Specific comments:

1. Nano-bridges form during solidification due to the formation of the B2 phase at the cell boundary. During severe plastic deformation, dislocations can traverse these nano-bridges, connecting the B2 precipitates on the cell boundaries. If we consider functionality solely in terms of the dislocation path, thicker bridges between two cells might be advantageous. So, why should our focus be on the nano-bridge?

Response: In the nano-bridged honeycomb microstructure, the thick B2 precipitates on the cell boundaries restrict the movement of dislocations (Fig. 3 of main text and Fig. 19 of supplementary materials). Concurrently, the breaking down of the nano-bridge of dislocations connecting the B2 precipitates facilitates the movement of dislocation away from the crack tip during severe plastic deformation (Fig. 3D). Furthermore, the nano-bridge of the dislocations provides resistance to dislocation movement in the initial phase of deformation which results in improved strength. Therefore, we argue that the thick B2 phase and the nano-bridge connecting the precipitates are both essential for the unprecedented strength and fracture resistance of nano-bridged honeycomb microstructure.

To clarify, the dissipation of plastic deformation away from the crack tip is the primary reason behind the high fracture toughness of ductile materials. However, commercially pure metals such as Cu, Ni and Al are very ductile, but their fracture toughness is low. Primarily because for high fracture toughness, dislocation needs to move away from the crack tip, but their movement needs to be strenuous to avoid strain localization and incur additional energy to open the crack tip. Both of these functions are important for high fracture resistance. Therefore, the fracture toughness (K_{Jc}) scales with the square root of yield strength and elastic modulus according to the micromechanics models of ductile fracture¹. We have added this discussion to the revised manuscript.

2. For fabrication, HT3 was used on the C(T) specimen to evaluate fracture toughness. How did microstructural evolution take place during heat treatment at 750 °C? How does this differ from the heat treatment carried out at lower temperatures (650 and 700 °C) over an extended period? The microstructure comprises B2 precipitates of various sizes, uniformly distributed. How were the nanoscale B2 precipitates uniformly distributed in the microstructure after the heat treatment (HT3)? Why were more nano bridges formed during HT3? A thermodynamic explanation is needed.

Response: We performed additional scanning electron microscopy to investigate the microstructure evolution after the four heat treatments. Figs. 8 and 9 of the supplementary materials show the back-scattered electron images of heat-treated samples. After all the heat treatments, the B2 phase on the cell

boundaries remains largely intact (Fig. 8). However, depending on the initial size and shape of the cellular structure inside a grain, they lose their shape in certain regions (Fig. 10D of the supplementary materials). The *fcc* matrix (cell interior) is supersaturated in Al (11.6 At%) due to rapid quenching during the laser powder bed fusion process (Fig. 23 of supplementary materials). Therefore, needle-shaped B2 precipitates form in the supersaturated *fcc* matrix, *i.e.*, the interior of the cells (Fig. 9). The microstructure in Fig. 9 shows that the heat treatment at 700 °C for 1 hour (HT1) is not sufficient for the growth of the B2 precipitates. They are barely resolved by SEM (Fig. 9A). After heat treatment at 700 °C for 2 hours (HT2), the fine B2 precipitates can be observed dispersed throughout the interior of the cells. When the heat-treatment temperature increases to 750 °C for 1 hour (HT3), the B2 precipitate needles grow longer and thicker than 700 °C for 2 hours (HT2). The temperature of 750 °C is also sufficient for homogeneous nucleation of the precipitates, leading to uniformly distributed nanoscale B2 precipitates. We do not think additional nano-bridges formed during the heat treatment (HT3).

Moreover, after 20 hours of annealing at 650 °C (HT4), the B2 precipitates formed in the *fcc* matrix are oval-shaped and much coarser than all other heat treatments. However, the 650 °C temperature was insufficient to generate a driving force for homogeneous nucleation of the precipitates. There is a bi-modal distribution of B2 precipitates after HT4 heat treatments, where smaller precipitates are not resolved by SEM (TEM image from ongoing study shown in the answer to the following comment). We have added part of this discussion to the revised manuscript.

3. At 298 K, the strength of HT4 is the highest, but at 77 K, that of HT3 is the highest. How do the microstructures of each differ? Why does HT3 have the highest strength at 77 K? If it's attributed to the formation of nano-bridges, please provide a quantified analysis for that.

Response: We discussed the microstructure of the specimens after heat treatments in response to the previous comment of the Reviewer. The yield strength of HT4 specimens is highest at both 298 K and 77 K. From the microstructure observed in Figs. 9C and D, the B2 precipitates are coarser after heat-treatment HT4 compared to HT3. A relatively finer and uniformly distributed precipitate in the HT3 microstructure should strengthen more than the HT4 microstructure. However, our TEM investigation of HT4 microstructure (Fig. i from ongoing study shown below) indicates a bi-modal distribution of precipitates where SEM does not resolve the smaller precipitates in Fig. 9D. The smaller precipitates with narrower spacing in HT4 microstructure impart them with highest strength. Therefore, it is clear that the size and shape of the B2 precipitates inside the *fcc* matrix control the strength of the heat-treated samples rather than the nano-bridges on the cell boundary. The ultimate tensile strength of the HT3 microstructure is higher at 77 K compared to all other cases. The difference in the ultimate tensile strength arises from the strain hardenability of the materials. However, since none of the heat-treated samples necked before fracture, it is challenging to comment on the strain hardenability of the individual microstructure.

Figure i. TEM bright field image showing bimodal distribution of B2 precipitates within *fcc* matrix after heat-treatment HT4.

4. In the supplementary materials, Fig. 7d, two types of B2 precipitates are present: one is the whisker-type nano-precipitates, and the other is the rounded (sometimes elliptical) micro-precipitates. How do the roles of each phase differ?

Response: The rounded B2 phase in Figure 7d (Figure 10D of the revised supplementary materials file) is the remnant of the cellular structure in the as-built condition. The secondary whisker-type nano-precipitates form during the heat treatment in the supersaturated *fcc* matrix. During deformation, smaller precipitates with a narrower spacing strengthen more effectively than coarser precipitates. Since the fraction of the whisker-type nano-precipitates is much higher than the coarse rounded precipitates, they are primary contributors to strengthening.

5. It seems that the B2 phase negatively impacts mechanical properties. However, without the B2 phase, nano-bridges cannot exist in the same way. Thus, the role of the B2 phase should be detailed. In Fig. 3, the deformation-induced nanotwins are obstructed by the B2 phase at the cell boundary. What role does the B2 phase play in fracture toughness? What is the optimal volume fraction, size, distribution, and morphology of the B2 phase to enhance both yield strength and fracture toughness?

Response: We believe the B2 phase on the cell boundaries does not adversely affect the mechanical properties. Instead, the combination of the thick B2 phase and the nano-bridge improves both the strength and fracture toughness. In the initial stage of deformation, the combination of the B2 phase on the cell boundaries and the nano-bridge of dislocations resist dislocation movement, enhancing the material's strength at 298 K and 77 K (Fig. 3). Moreover, during severe plastic deformation, the nano-bridge of dislocations breaks down and allows the movement of dislocation away from the crack tip—the combination of these results in an exceptional combination of strength and fracture toughness. We have added additional TEM images from the plastic zone near the crack tip to highlight the role of the B2 phase in blocking dislocation movement at 298 K (Figs. 19A and B of the supplementary materials) and nano twins at 77 K (Figs. 19C and D of the Supplementary Materials).

Conventional ways of strengthening and toughening materials involve manipulating the size and morphology of grains and precipitates. In the present study, the architecture of the material's microstructure was tailored by additive manufacturing to achieve simultaneous strengthening and toughening. Apart from the size and morphology of the B2 precipitates, the whole 3D honeycomb structure is essential for unprecedented fracture resistance of the material. Since altering the volume fraction, size, distribution, and morphology of B2 precipitate is impossible without affecting the overall 3-D honeycomb architecture (including the nano-bridge of dislocation), it is difficult to comment on the Reviewer's question about optimizing these features for improving the toughness without considering the 3D architecture of the microstructure. This is an interesting point but we will have to leave this question open for future studies.

6. In the Supplementary Materials Fig. 20, APT analysis was conducted on the *fcc* cell, intercellular B2, and the BCC phases. Nanotwins and dislocations are primarily obstructed by the B2 phase but traverse through the nanotwins, which is advantageous for enhancing toughness. The high dislocation density is thoroughly discussed in numerous figures, but I'm intrigued by the APT results at the boundary between the nano-bridge and the B2 phase. If the authors can clarify dislocation behavior in terms of atomic distributions, it would bolster the paper's argument by elucidating the channeling effect of nano-bridges.

Response: Thanks for the inspiring comments. APT analysis has become a standard materials characterization method for phase identification, interface segregation or quantification of nanoclusters in metallurgy. This work used APT to characterize the multiphase HEA microstructure combining with TEM, *i.e.*, FCC, B2, and BCC phases. In particular, the critical strengthening phase of the B2 (NiAl) and the spinodal decomposed BCC (CrCoFe) particles (~10 nm) were unambiguously identified using the APT technique (Fig. 1 of the main text and Fig. 5 of the supplementary materials). However, using APT, it is still technically challenging to detect and characterize dislocations and nanotwins in the deformed microstructure. As the APT volume is reconstructed after layer-by-layer atomic evaporation

via software, during which the accurate atomic configurations in 3D space might be partly lost. Therefore, at the moment, we cannot reveal dislocation behavior reliably in terms of atomic distributions using APT. Fortunately, the combination of high-resolution TEM and EBSD (Fig. 3 of main text and Figs. 4, 17-20 of supplementary materials) enables us to show an in-depth understanding of the "channelling effects" of nano-bridges during plastic deformation, *i.e.*, dislocation movement restricted by B2 phase and allowing their movement through to the nano-bridge to improve the resistance to crack propagation.

7. Minor comments:

(a) Even though the authors reference ref. 25, the explanation for Supplementary Materials Figure 3 is insufficient. Please indicate the zone axis in the TEM DP and provide a detailed explanation of the ordered B2 and disordered A2 phase.

Response: We revised the manuscript to clarify the spinodal decomposition process and the presence of A2/B2 phases in the microstructure. We have added the zone axis information to all the TEM diffraction patterns in the revised manuscript.

(b) Line 436-439: In refs. 4 and 5, please remove "(1979)".

Response: Thank you for pointing out this error. We have removed "(1979)" in the revised manuscript.

(c) Line 211: Please change "Figure" to "Fig."

Response: Corrected in the revised manuscript.

(d) Supplementary Materials Figures 18, 19, and 20 lack explanations in the manuscript.

Response: Supplementary Figure 18 (Figure 12 of revised supplementary materials) is an enlarged part of Figure 2 of the main text. Similarly, Figures 19 and 20 (Figures 5 and 23 of the revised supplementary materials) are part of the main text's APT result shown in Figure 1E. In the revised manuscript, we have discussed these figures.

Reviewer #2:

General comment: The authors present a novel and interesting study, wherein the Al_{0.5}CrCoFeNi high-entropy alloy (HEA) develops a unique microstructure due to its fabrication via laser-beam powder bed fusion process. Many interesting aspects regarding the formation of BCC-FCC microstructure and the corresponding enhancement in strength-ductility/toughness combination are explained in depth. It is well written, and thus suggested to be accepted after the following comments are addressed:

Response: We thank the Reviewer for their positive comment. We have addressed the specific comments below and made changes to the manuscript accordingly.

Specific comments:

1. The distinction between the A2 and B2 phases is important in the Al_{0.5}CrCoFeNi HEA system and shall be elaborated further. How do the authors clearly distinguish between the ordered and disordered BCC phases via the TEM investigations (supplementary materials Figure 3) and EBSD (Fig. 1c)?

Response: We agree with the Reviewer's assessment that the distinction between A2 and B2 phases is crucial for this manuscript. Therefore, in addition to TEM, we carried out an APT line scan of the prior B2 phase on the cell boundary, which goes through spinodal decomposition, forming secondary ordered B2 and disordered A2 phases (Supplementary Materials Figures 5). In the revised manuscript, we have clarified the differences between the A2 and B2 phases based on these APT results.

2. The authors mention "honeycomb" (in several instances, for example, Line 115) microstructure in reference to the cell boundaries of the as-printed alloy. The cellular structures only have a typical irregular hexagonal shape (M. Moneghan, C. Williams, R. Mirzaeifar, J. Mater. Res. 2020, 35, p.1984)

and the presence of strengthening advantages from the naturally occurring "honeycomb" structure is debatable in the 3D-printed HEA.

Response: The manuscript indicates that cellular microstructure resembles a honeycomb structure. We fully agree with the Reviewer that the cellular structure is of typical irregular hexagonal shape. However, as shown in Figure 1A, the columnar nature of these cells resembles a honeycomb structure. Also, we do not claim strengthening advantages parallel to naturally occurring "honeycomb" structures. The strengthening comes from cellular structure and high dislocation density, including the nano-bridge connecting the B2 phase. We appreciate the reviewer bringing our attention to the referred paper², where the strengthening from cellular structure and the importance of rupturing cellular boundaries to spread the damage are highlighted. A parallel can be drawn here with breaking the dislocation nano-bridge in the later stage of the plastic deformation to spread the damage in the material and avoid strain localization. We have added a part of this discussion to the revised manuscript.

3. The as-built microstructure has been termed as "multiphase" (Line 244). Unless the A2 and B2 phases are clearly demarcated and their role in deformation is explicated, the multiphase nature of the HEA may be questionable.

Response: We have revised the manuscript to demarcate between the A2 and B2 phases. The combination of APT and TEM microscopy results was used to identify these phases (Figure 1 of the main text and Figures 4 and 5 of the supplementary materials). During the printing process, the B2 phase on the cell boundaries goes through solid-state spinodal decomposition, forming ordered secondary B2 and disordered A2 phase. The spinodal mixture of *bcc* phases, referred to as the prior B2 phase in the manuscript, is present on the cell boundaries and restricts the movement of dislocations/nano-twins to strengthen the material. Concurrently, the nano-bridge of dislocations connecting the prior B2 phase on the cell boundaries breaks down under severe plastic deformation to create a highway for the movement of dislocations away from the crack tip. This discussion has been added to the revised manuscript (Figure 3 of the main text and Figure 19 of revised supplementary materials).

4. The 3D-printed metallic structures are well known to be anisotropic, wherein the large columnar grains grow along the building direction. However, it is interesting to note that even the cellular structure (which is generally randomly oriented, Y.-K. Kim, S. Yang, K.-A. Lee, Addit. Manuf. 2020, 36, 101543) in the current study (Fig.1a) seems to be anisotropic and elongated along the building direction. The related explanations on this will add significant value to the microstructural discussion.

Response: We thank the Reviewer for the careful observation. We want to point out that columnar grains are prevalent in additively manufactured materials. The columnar grains are a consequence of columnar growth of the cellular structure. In the article referred to by the reviewer³, the grains are not columnar (at least on the TD plane). However, Fig. 4b of this article shows columnar cells in the SD plane. The grain orientation and morphology are intrinsically linked with the orientation and the morphology of the cells⁴. Therefore, the columnar grains are always associated with columnar cells since one is a consequence of the other⁵. We have added this explanation to the revised manuscript.

5. The building and scanning directions shall be indicated in Figure 3.

Response: Thank you for pointing this out; we have addressed this in the revised manuscript.

6. What is the equilibrium partition coefficient of Ni and Al, and how does it compare with those of Cr, Co, and Ni, such that the segregation of Ni and Al took place in the cell boundaries? Can the authors indicate what factors influence such segregation and the equilibrium partition coefficient during laser beam powder bed fusion?

Response: We used the Thermo-Calc software with the latest TCHEA6 database to compute the equilibrium partition coefficients of the five components with the Scheil-Gulliver model in Al_{0.5}CrCoFeNi HEA. It is shown that the equilibrium partition coefficients of Ni and Al are ~0.98 and ~0.81, and those of Cr, Co and Fe are ~1.01, ~1.12, and ~1.01, respectively. Its value below unity means

that the element partitions more to cell boundaries, and the more it deviates from unity, the more severe the partitioning behavior. Moreover, the dramatic changes in the partition coefficients are due to the B2 (NiAl) phase precipitation at cellular boundaries. The cooling rate is the primary factor influencing segregation or partitioning behavior. Due to the cooling rate of $\sim 10^5$ - 10^7 K/s of the L-PBF process, the highest among all other powder-bed fusion and directed energy deposition (DED) AM processes, its cell structure is finest, and the elemental segregation is relatively highest, leading to the formation of B2 precipitates at cell boundaries. These discussions and the thermodynamic calculations (Fig. 3 of supplementary materials) have been added to the revised manuscript.

Supplementary Materials Figure 3. Thermocalculation of equilibrium partition coefficients of the five components (Al, Cr, Co, Fe and Ni) with the Scheil-Gulliver model in $Al_{0.5}CrCoFeNi$ HEA.

7. It is difficult to state whether the crack growth is chiefly along the cell boundaries or through the cell interiors. The cell interiors consist of a more ductile FCC phase whereas the harder BCC phase is present on the cell boundaries. The deformation mechanism explained on page 8 (Lines 197-208) could be elaborated further in this respect.

Response: Even very close to the crack tip, the strain localization in ductile *fcc* cells is not high enough to initiate debonding of the hard *bcc* phase on the cell boundaries. We can state this explicitly because no secondary crack is present near the crack tip (Figs. 3A, B, and C). Lack of secondary cracks highlights the importance of the size and morphology of the honeycomb cellular structure, including the dislocation nano-bridge, which facilitates the movement of dislocation away from the crack tip. We have revised our discussion on page 8 to clarify this.

8. Supplementary Materials Figure 18. Why the data points first move towards left (i.e., the crack extension ($\Delta(a)$) decreases or the crack becomes shortened) with increasing J-integral value needs to be explained.

Response: What the Reviewer has pointed out is referred to as the swayback phenomena in ASTM E1820⁶, characterized by a decrease in crack mouth opening compliance when force is applied to a C(T) specimen, resulting in an apparent crack de-extension determined using compliance calculations. As the sample opens up during fracture toughness tests, the relative rotation of pin/sample and pin/fixture is opposite. Swayback is commonly observed when friction between the pin and the sample/fixture (the pin used to load the C(T) sample shown in Fig. 21C of the supplementary materials) results in slippage of the C(T) sample, giving a false indication of crack closure. To minimize errors caused by friction, the fixture and pin are designed so that the pin rolls on the fixture without slipping and maintains no relative motion between the pin and sample hole; thus, frictional forces minimally affect the results. This approach has minimized friction-induced errors, as evidenced by the absence of swayback in most of our *J-R* curves.

Nevertheless, when the sample becomes harder (after heat treatment), yield strength (YS) approaching that of peak-aged maraging steel—the material of our pin—the indentation/brinelling of the pin onto

the C(T) specimen hole becomes insufficient, leading to slippage at low force. However, as the force increases, the indentation/brinelling intensifies, and the frictional force also increases to the point where there is no relative motion between the sample and the pin. Once this relative motion ceases, the friction-induced compliance error is minimized. Therefore, the swayback is only observed at low forces and does not affect the fracture toughness measurements. ASTM recommends removing the negative crack extension data if the reversal in the crack extension exceeds 0.5 mm⁶; however, the swayback in our present result is much less; therefore, we did not remove these data points from the *J-R* curve. We have added a part of this discussion to the caption of Fig. 12 of the Supplementary Materials in the revised manuscript.

References

1. Ritchie, R. O. & Thompson, A. W. On macroscopic and microscopic analyses for crack initiation and crack growth toughness in ductile alloys. *Metallurgical Transactions A* **16**, 233–248 (1985).
2. Moneghan, M., Williams, C. & Mirzaeifar, R. Deformation mechanisms and defect tolerance in the microstructure of 3D-printed alloys. *J Mater Res* **35**, 1984–1997 (2020).
3. Kim, Y. K., Yang, S. & Lee, K. A. Compressive creep behavior of selective laser melted CoCrFeMnNi high-entropy alloy strengthened by in-situ formation of nano-oxides. *Addit Manuf* **36**, (2020).
4. Sofinowski, K. A., Raman, S., Wang, X., Gaskey, B. & Seita, M. Layer-wise engineering of grain orientation (LEGO) in laser powder bed fusion of stainless steel 316L. *Addit Manuf* **38**, 101809 (2021).
5. Becker, T. H., Kumar, P. & Ramamurty, U. Fracture and fatigue in additively manufactured metals. *Acta Mater* **219**, 117240 (2021).
6. ASTM Standard E1820. Standard test method for measurement of fracture toughness. *ASTM Book of Standards* 1–54 (2013) doi:10.1520/E1820-13.Copyright.

REVIEWER COMMENTS

Reviewer #1 (Remarks to the Author):

The authors have addressed most of my questions, yet I still wonder if the authors can control the thickness of the bridges connecting the B2 precipitates. Considering that these bridges potentially aid in the movement of dislocations away from the crack tip during severe plastic deformation, the thickness of these bridges could be a crucial factor influencing the material's fracture toughness. Since the role of nano-bridges in toughening is the most crucial part of this paper, direct and sufficient evidence is needed to demonstrate that nano-bridges facilitate the movement of dislocations away from the crack tip.

Reviewer #2 (Remarks to the Author):

The authors have addressed all the reviewers' comments properly in the manuscript. I therefore recommend to publish the manuscript in Nature Communications.

Responses to the Reviewers' comments

We thank the reviewers for carefully reading our revised manuscript and providing comments. We respond to the comments below (our responses are in blue.) We also highlight the changes made to the text in red.

Reviewer #1: The authors have addressed most of my questions, yet I still wonder if the authors can control the thickness of the bridges connecting the B2 precipitates. Considering that these bridges potentially aid in the movement of dislocations away from the crack tip during severe plastic deformation, the thickness of these bridges could be a crucial factor influencing the material's fracture toughness. Since the role of nano-bridges in toughening is the most crucial part of this paper, direct and sufficient evidence is needed to demonstrate that nano-bridges facilitate the movement of dislocations away from the crack tip.

Response: We thank the Reviewer for going through the revisions. The thickness of the nano-bridge of dislocations connecting the B2 phase is crucial for the material's fracture toughness. During solidification, as mentioned in the manuscript, the solutes Ni and Al segregate on the cell boundaries, forming B2 precipitates preferably near triple points where the flux of solute segregation is highest. Although the solutes also segregate in other regions of the cell boundaries, the concentration of Ni and Al is not high enough in these regions to form the B2 phase. However, these solutes on the cell boundaries trap the dislocations nucleated by thermal cycling, forming dislocation nano-bridges that connect the B2 phase on the cell boundaries. Since the nano-bridge of dislocations is an indirect result of solute segregation on the cell boundaries, we can speculate that the amount of solute segregation would be important in controlling their thickness. The morphology of similar dislocation structures on the cell boundaries of laser powder bed fusion processed 316L SS has been reported to be altered by changing the cooling rate, the thermal gradient in the melt pool, and the number of heating and cooling cycles for each layer^{1,2}. However, for the high entropy alloy Al_{0.5}CrCoFeNi, these parameters will affect both the morphology of the B2 phase and the dislocation nano-bridge on the cell boundaries; therefore, a detailed investigation is required to determine the primary factors controlling the thickness of the dislocation nano-bridge. Moreover, as discussed in the paper, the breakdown of the nano-bridge is essential to facilitate the movement of dislocations further away from the crack tip. Therefore, a thicker nano-bridge that doesn't allow the movement of the dislocations away from the crack tip would be detrimental to the fracture resistance.

We agree that direct evidence of nano-bridge breakdown and dislocation moving through those bridges is very important to support our proposed deformation mechanism. Accordingly, Fig. 3 of the manuscript shows evidence demonstrating how the nano-bridges between the B2 phases facilitate movement of dislocations away from the crack tip. The TEM images in Fig. 1D show proof of these nano-bridges on the cell boundaries. Figs. 3A, B and C show dislocations breaking these nano-bridges and moving away from the crack tip during fracture at 77 K. Figs. 3B and C show dislocations bands between the B2 phases on the cell boundaries where the nano-bridges were present before the plastic deformation. The TEM image in Fig. 3D shows dislocations moving through the nano-bridges (red arrows) between the B2 phases during fracture at 298 K.

We have revised the manuscript to clarify these points further.

Reviewer #2: The authors have addressed all the reviewers' comments properly in the manuscript. I therefore recommend to publish the manuscript in Nature Communications.

Response: We thank the Reviewer for their time and effort in reviewing the manuscript.

References:

1. Bertsch, K. M., Meric de Bellefon, G., Kuehl, B. & Thoma, D. J. Origin of dislocation structures in an additively manufactured austenitic stainless steel 316L. *Acta Mater* **199**, 19–33 (2020).

2. Becker, T. H., Kumar, P. & Ramamurty, U. Fracture and fatigue in additively manufactured metals. *Acta Mater* **219**, 117240 (2021).

REVIEWERS' COMMENTS

Reviewer #1 (Remarks to the Author):

The reviewer extends gratitude to the authors for their thoughtful and comprehensive responses. I am now pleased with the revisions made to the manuscript and would like to endorse its publication in this journal.